# Leading Dynamical Processes of Global Marine Heatwaves in an Ocean State Estimate

Jacopo Sala<sup>1</sup>, Donata Giglio<sup>1</sup>, Antonietta Capotondi<sup>2</sup>, Thea Sukianto<sup>3</sup>, and Mikael Kuusela<sup>3</sup>

Correspondence: Jacopo Sala (jacopo.sala@colorado.edu)

**Abstract.** Marine heatwaves (MHWs) have emerged as a very active area of research due to the devastating impacts of these events on marine ecosystems across different trophic levels. Yet, a clear understanding of the local drivers of these extreme ocean conditions is still limited at a global scale. Observations of the terms needed to constrain ocean heat budgets are very sparse, ocean reanalysis products are generally non-conservative and inadequate to conduct accurate heat budget analyses, and the fidelity of climate models in simulating MHWs is still unclear. In this study, we make use of Argo floats observations, a satellite-based sea surface temperature product, and the Estimating the Circulation and Climate of the Ocean (ECCO) state estimate to assess MHW characteristics over the global ocean. ECCO is then used to evaluate local MHW drivers. ECCO assimilates observations using the adjoint methodology, which optimizes the system trajectory given the observational constraints in a conservative fashion, making it an ideal product for the estimation of heat budgets. The representation of MHWs in ECCO is overall consistent with observations, although ECCO tends to underestimate MHW frequency and intensity and overestimate duration, relative to the observational products. Atmospheric forcing emerges as the dominant contributor to MHW onset and decline across most regions, while ocean dynamics, including advective and diffusive convergence of heat, play crucial roles in the equatorial regions, specific extra-tropical zones (e.g., western boundary currents such as the Gulf Stream and Kuroshio), and the Southern Ocean. Regional analyses in the Northeast Pacific, Southwest Pacific, and Tasman Sea, show diversity in leading dynamical mechanisms for MHW onset and decline both across regions and across events in the same regions: while air-sea exchanges of heat may contribute most frequently to MHW onset and decline, other mechanisms can also often provide dominant contributions and at times be the main driver. A more complete understanding of MHWs and their drivers is crucial for predicting their initiation, duration, intensity and decline, to ultimately inform the development of mitigation and adaptation strategies for affected communities.

#### 20 1 Introduction

Events of extreme warming in the ocean have received increasing attention due to their profound ecological and socioeconomic impacts (Smith et al., 2021), as they are often associated with widespread ecosystem disruptions (Guo et al., 2022; Smith et al., 2023), coral bleaching (Brown, 1997; Donovan et al., 2021), shifts in species distributions (Lonhart et al., 2019), and changes in oceanic primary productivity (Frölicher and Laufkötter, 2018; Pearce et al., 2011). A better understanding of these extreme

<sup>&</sup>lt;sup>1</sup>University of Colorado - Boulder, Department of Atmospheric and Oceanic Sciences, Boulder, CO, USA

<sup>&</sup>lt;sup>2</sup>NOAA Physical Sciences Laboratory, CIRES, University of Colorado - Boulder, Boulder, CO, USA

<sup>&</sup>lt;sup>3</sup>Carnegie Mellon University, Department of Statistics & Data Science, Pittsburgh, PA, USA

events and how they will affect marine ecosystems and livelihoods in the future is crucial to help coastal communities adapt to

Extreme warm conditions associated with processes internal to the climate system occur in the presence of a long-term global warming trend (Figure 1), which can increase their frequency of occurrence and intensity, and exacerbate their impacts. While atmospheric greenhouse gas concentrations are nearly spatially uniform, the long-term warming of the ocean surface shows regional patterns that affect the atmospheric circulation, regional changes in rainfall as well as the global climate sensitivity (Xie, 2020). Both the distribution of radiative forcing (greenhouse gas vs. aerosols concentrations) and the ocean circulation (e.g., global overturning circulation, wind driven circulation and upwelling) shape observed patterns (Xie, 2020), leading, for example, to long-term warming signals in the area of the western boundary currents (Wu et al. (2012); Figure 1). The presence of this large sea surface temperature (SST) trend has led to discussions on appropriate definitions of MHWs in a changing climate (Amaya et al., 2023; Burrows et al., 2023). Specifically, MHW definitions that do not factor out the effects of having a trend in the data, e.g., those based on a fixed baseline to define the climatology used to assess extremes (Hobday et al., 2016) lead to an increase in frequency and intensity of these extreme events (Seneviratne et al., 2021; Wigley, 2009; Evans et al., 2020; Oliver et al., 2018; Oliver, 2019; Smale et al., 2019; Scannell et al., 2016; Xu et al., 2022), and may also result in permanent MHW conditions in region of accelerated warming (Capotondi et al., 2024). The inclusion of the long-term trend in MHW definitions can also be expected to confound our understanding of the processes leading to MHW development, and their associated predictability (Wulff et al., 2022). On the other hand, impact studies may need to consider the total SST anomalies, especially when concerned with marine species that have a slow adaptation timescale (Smith et al., 2024). Implications of different MHW definitions, and their suitability for different applications, are discussed in depth in Smith et al. (2024).

In this paper, we aim at understanding the processes underpinning MHW events arising from internal climate processes, due, e.g., to an increase in variance (Xu et al., 2022). In particular, variability associated with the El Niño Southern Oscillation appears to increase the likelihood and/or persistence of MHWs in different parts of the world (Frölicher and Laufkötter, 2018; Oliver et al., 2018; Sen Gupta et al., 2020; Xu et al., 2021; Capotondi et al., 2022; Gregory et al., 2024a), suggesting that changes related to ENSO could account for a substantial portion of the future increase in MHWs (Deser et al., 2024; Capotondi et al., 2024). A better understanding of the different drivers of MHWs in different regions of the ocean can help improve MHWs forecasts and the information policymakers use to develop proactive mitigating strategies (Holbrook et al., 2019, 2020). Local atmospheric processes and oceanic circulation patterns play critical roles in shaping the dynamics of MHWs, and their intensity and duration (Pujol et al., 2022; Marin et al., 2022). Teleconnections from large-scale modes of variability can play an important role in modulating the local drivers, as seen in events like the 2013-2015 Northeast Pacific MHW, known as "The Blob" (Di Lorenzo and Mantua, 2016).

While a large fraction of the literature has focused on the development phase of MHWs, the processes responsible for the demise of these events are also important. Thus, here we consider drivers of both the onset and decline phases of MHW events (Figure 3) during 1992-2017. For this purpose, we use, for the first time in MHW studies, the Estimating the Circulation and Climate of the Ocean (ECCO) state estimate. This dynamically consistent ocean state estimate incorporates a wide range of oceanic and atmospheric observations (Forget et al., 2015) and provides daily temperature fields, as well as all heat budget

terms computed at each model time step, ensuring closure of the heat budget. We note that the attribution of heat budget terms, particularly the separation of advective and diffusive contributions, is inherently tied to model resolution, and coarse-resolution products may misattribute unresolved advective processes to subgrid-scale diffusion. Yet, while eddy-permitting (0.25°) or eddy-rich (0.1°) models improve MHW realism in highly dynamic regions (e.g., western boundary currents), even coarserresolution models can qualitatively capture large-scale MHW patterns in less active regions (Pilo et al., 2019). While ECCO's 1x1 degree resolution is a limitation, results from (observations constrained) ECCO's results provide an excellent complement to results from free-running models (of comparable resolution) 4 described in recent MHW studies, such as work using the CESM large ensemble (1°-1.5°: Deser et al. (2024)) and an analysis based on the GFDL ESM2M coupled Earth System Model (with a nominal 1° resolution increasing (in longitude) to 1/3° near the equator; Vogt et al. (2022), describing MHW dynamics and local drivers of MHW onset and decline phases in different seasons, over a 500-year period). ECCO provides an optimal balance between resolution, dynamical fidelity (thanks to being constrained by observations), and spatial coverage for global, process-based MHW analysis. Also, compared to other data assimilating products, ECCO enables heat budget closure—an essential advantage for our process-focused analysis. Our goals are to (1) describe MHW characteristics in ECCO relative to satellite-based and in-situ observations, and (2) investigate the roles of oceanic and atmospheric drivers of MHWs over the global ocean, providing valuable insights for ecosystem management, climate adaptation strategies, and sustainable resource planning. Data and methods used in our analysis are described in Section 2 and 3, respectively. Results and discussion are presented in Section 4. Conclusions are drawn in Section 5.

## 2 Data

80

# 2.1 ECCO v4r4 Ocean State Estimate

We use data from Version 4 Release 4 (v4r4) of the ECCO (Estimating the Circulation and Climate of the Ocean) ocean reanalysis (Consortium et al., 2021; Forget et al., 2015) from January 1992 to December 2017. ECCO v4r4 is a conservative and dynamically consistent reanalysis based on the MIT general circulation model (Marshall et al., 1997) configured with 50 vertical levels and with a horizontal resolution of 1° throughout the globe. The ECCO ocean state estimate includes observational data used to constrain the model, e.g., from Argo floats, moorings, ship-based measurements (e.g., CTDs) and satellites. We use ECCO potential temperature, salinity, heat and freshwater fluxes fields, and both daily and monthly output to examine how the definition of MHW characteristics is influenced by the temporal resolution of the data used.

# 2.2 Argo OHC

We use monthly ocean heat content (OHC) fields (for the layer 15-50 dbar) based on Argo profile data (Wong et al., 2020). In Giglio et al. (2024), Argo observations are mapped on a  $1 \times 1$  degree grid using locally stationary Gaussian processes defined over space and time, with data-driven decorrelation scales (Kuusela and Stein, 2018). Also, a linear time trend is included in

**Figure 1.** Linear trend in (a) sea surface temperature based on the NOAA OISST V2 data product and (b) near-surface temperature from the ECCO v4r4 ocean state estimate (°C/decade). The trend is estimated from monthly data over the period 1992-2017. Panel a also shows regions of interest in our analysis: (box 1) the Northeast Pacific, (2) the Southwest Pacific and (3) the Tasman Sea.

the estimate of the mean field, along with spatial terms and harmonics for the annual cycle. We use mapped fields for the period 2004-2017, which overlaps with the monthly ECCO data.

# 2.3 NOAA OISST V2

We use sea surface temperature data from the NOAA Optimum Interpolation Sea Surface Temperature (OISST) V2 High Resolution Dataset at a 1/4° resolution (Huang et al., 2021) for the period 1992 to 2017. This product has been extensively used by previous researchers for MHW identification, and has the advantage of a high horizontal resolution and the availability of data at a daily timescale.

## 3 Methods

#### 3.1 Marine heatwave identification

As in Hobday et al. (2016), marine heatwaves are identified here based on deviations from a time-evolving "normal" sea surface temperature (or upper ocean heat content) that exceed a certain threshold over a specified period. This threshold is set as the seasonally varying 90th percentile, meaning that MHWs are characterized by sea surface temperature (or upper ocean heat content) anomalies that are warmer than 90% of anomalies in each season over a time frame of interest (for daily data, the seasonally varying 90th percentile is defined daily, using an 11-day moving window). Anomalies are defined with respect to the seasonal cycle and linear trend during the total duration of the period considered. Thus, unlike (Hobday et al., 2016), who used the 1982-2012 fixed baseline, here we use the entire period for the definition of the climatology after removing the linear trend.

For MHWs to be identified, the anomalies in temperature (or ocean heat content) need to exceed the selected threshold for more than a minimum duration, to ensure that short-lived fluctuations in temperature or ocean heat content are not categorized as MHWs and that only long-lived thermal anomalies are included in the analysis. The minimum duration of MHW events is set based on the time resolution of each dataset, with 1 month for monthly data and 5 days for daily data. For daily data, the end of the event corresponds to anomalies remaining below the threshold for more than 2 days (after the initial 5 days of continuous anomalies above the threshold), i.e., we treat two MHW events separated by less than 2 days as a single, continuous MHW event. This approach is consistent with Hobday et al. (2016) and prevents the misclassification of prolonged MHWs as multiple shorter events, allowing us to maintain the continuity of the thermal anomaly and better reflect the actual duration and intensity of the events in our analysis.

Once the percentiles are calculated and the MHW thresholds are applied, we can identify the start and end dates of each event, track their evolution, and analyze their duration. Additionally, when using daily data, we can identify the MHW onset versus decline phase, with the onset phase going from the start of the event to the peak intensity (included) and the decline starting just after the peak intensity and going to the end of the event (Figure 2). By explicitly defining different phases we can study processes driving MHW intensification and dissipation.

Finally, using daily data, we compare the statistics of MHWs of different durations, i.e., MHWs lasting between 5 and 29 days, and events lasting 30 days or longer. In addition, since some previous studies have utilized monthly averaged data, especially in the case of long-lasting events like the Blob (Xu et al., 2022; Capotondi et al., 2022), here we also compare the statistics obtained using monthly averages with those derived from daily data.

## 125 3.2 ECCO ocean heat budget

To compute the global ocean heat budget in the upper ocean (5–55 meters), we use the ECCO v4r4 product at daily and monthly resolutions, following the methodology outlined in Forget et al. (2015). The evolution of temperature follows:

Figure 2. Schematic representing how we define and identify MHWs and their onset (red) and decline (blue) phase.

$$\frac{\partial \theta}{\partial t} = -\nabla \cdot (\theta \mathbf{u}) - \nabla \cdot \mathbf{F}_{\text{diff}}^{\theta} + \mathbf{F}_{\text{forc}}^{\theta}$$
(1)

where:

- $-\frac{\partial \theta}{\partial t}$  is the rate of change of potential temperature,
- $-\nabla \cdot (\theta \mathbf{u})$  is the advective divergence term (encompassing both horizontal and vertical components of heat transport),
- $-\nabla\cdot\mathbf{F}_{\mathrm{diff}}^{\theta}$  is the diffusive divergence term,
- $-\mathbf{F}_{\mathrm{forc}}^{\theta}$  denotes the atmospheric forcing term, including surface heat fluxes.

Ocean heat budget terms follow from the terms in this equation as ocean heat content is the volume integral of  $\theta \cdot \rho \cdot c_p$ , where density  $\rho$  is 1,030 kg/m<sup>3</sup> and specific heat  $c_p$  is 3,989.244 J/(kg·K), as described in McDougall and Barker (2011). We note that ECCO ocean heat budget closes practically exactly, as residuals are orders of magnitude smaller than the other budget terms. We compute composite fields for each heat budget term (atmospheric forcing, advective convergence, and diffusive convergence) separately during the onset and decline phases of marine heatwaves (MHWs), examining the relative contributions of these terms to ocean heating and cooling during the two phases (Figure 7). To assess how often each term contributes to warming during MHW onset and cooling during MHW decline, we compute the frequency of positive (average) contributions during onset and negative (average) contributions during decline (Figure 8). Furthermore, we evaluate how often each term is the largest or smallest contributor to the heat budget during each phase (Figures 9, 10), e.g., calculating the percentage of instances (across all the events) when a term is the largest contributor, however similar to other terms it may be.

Towards summarizing our global and regional findings for the leading dynamical processes driving MHWs, we introduce the terms "the" leading term and "a" leading term, defined as follows. For each phase, we identify the budget terms that contribute to it (among forcing, advective convergence, diffusive convergence), then we sort them by the magnitude of the contribution. A term provides "the" leading contribution if it exceeds the next largest term by at least 30%. A term also provides the leading contribution if it is the only process contributing 5 to the phase of interest. If the largest two contributors are both greater than the third by at least 30%, but neither is larger than the other by 30%, then each of the two terms provides "a" leading contribution. The same happens if these two terms are the only contributors and neither is larger than the other by 30%. If none of the terms provides a contribution that is 30% larger than other contributions, all three terms contribute comparably. We note that while the 30% threshold is arbitrary, it serves the purpose of summarizing our results, and visually identifying which processes are most often leading terms. Our findings are robust to  $\pm 5\%$  change in the (30%) threshold percentage used (not shown). In addition to the global analysis, we perform a regional analysis, averaging heat budget terms and ocean heat content anomalies in three regions of the Pacific Ocean: the Northeast Pacific (region 1 in Figure 1-a, NEP), the Southwest Pacific (region 2, SWP), and the Tasman Sea (region 3, TASMAN). We evaluate the cumulative sum of each heat budget term during the onset and decline phases of MHW events, to estimate their contributions to OHC changes (e.g., identifying "a" leading term versus "the" leading term), and show how the relative importance of the different processes varies across regions and events in the same region (Figures 11–13).

#### 160 4 Results and Discussion

155

165

# 4.1 Representation of MHWs in ECCO compared to observations

ECCO provides an overall good representation of the spatial patterns of both the long term linear trend in upper ocean temperature (e.g., Figure 1) and the 90th percentile anomalies (Figure S1 in the Supplement); also, spatial patterns of MHWs frequency, average duration, and average intensity in ECCO are consistent with observations (Figures 3-6). Yet, a smaller number of near surface MHW events shorter than a month (i. e. duration between 5 and 29 days) is seen in ECCO compared to observations (Figure 3a d), with only some of these events showing a signature in upper (5-55m) ocean heat content (Figure 3 g). Some of the differences between ECCO and observations may be related to ECCO's resolution: Pilo et al. (2019) show that while model configurations with resolutions ranging from 1° to 1/10°, can all qualitatively represent broad-scale global patterns of MHWs, modeled MHWs tend to be weaker, longer-lasting, and less frequent than in observations, especially for models with lower resolution. High-resolution, eddy-permitting models perform generally better in representing MHW characteristics, particularly in dynamic regions like western boundary currents, but still exhibit biases (Pilo et al., 2019). Discrepancies between models and observations are due in part to smoother SST time series and longer autocorrelation times (Cooper, 2017) in models (compared to observations), which can reduce short-lived variability and emphasize events of longer duration.

Despite the limitations highlighted above, ECCO provides a useful representation of MHWs in the global ocean. Also, the availability of ECCO daily fields allows us to study dynamical mechanisms of MHWs during the onset and decline phase of the

**Figure 3.** Total number of MHW events detected during the period 1992-2017, using (a-c) OISST V2 data, (d-f) ECCO v4r4 near-surface temperature, (g-i) ECCO v4r4 ocean heat content (5-55m). Panels a, b, d, e, g, h (left and middle columns) are based on daily data and include MHW events with a duration (a, d, g) within 5 and 29 days and (b, e, h) of 30+ days. Panels c, f, i (right column) are based on monthly data and include MHW events with a minimum duration of 1 month.

event. In the following, the analysis will focus on MHWs with a minimum duration of 5 days, without distinguishing between events lasting 5-29 days and those lasting 30 or more days, as results from the two categories are similar (not shown).

# 4.2 Leading dynamical processes for MHWs

In most of the ocean, the composite amplitude of atmospheric forcing during MHWs onset and decline is larger than the amplitude for advective and diffusive convergence (Figure 7b, c). The spatial distribution of this term is quasi-uniform, with the exception of the tropics. Atmospheric forcing contributes to warming during the onset phase and cooling during the decline (i.e., it is positive during the onset and negative during the decline phase) in most regions of the global ocean and for most events (Figure 8 and 9); yet, on average, atmospheric forcing cools the upper ocean during both the onset and decline phase near the equator (Figure 7b, c). In addition, it contributes to the decay most of the time (Figure 8a b), consistent with previous studies (Holbrook et al., 2019; Vogt et al., 2022). In equatorial regions, advective convergence plays a lead role in the onset and decline

Figure 4. As in Figure 3, now for the average duration of MHW events.

of MHW (and ENSO) warm anomalies (Figure 9c, d and 10 a-d), consistent with ENSO dynamics, e.g., Jin (1997); Capotondi (2013). On average, advective convergence contributes to both the onset and decline phase only near the equator (panel e, f in Figure 7), in the subtropics and western boundary currents (consistent with previous studies, (e.g., Li et al., 2022; Zhang et al., 2023) in the Southern Red Sea and Gulf of Aden (Nadimpalli et al., 2025), and along the Antarctic Circumpolar Current (Figure 7e, f). In these regions, the advective convergence is most often a contributor only to the onset phase (8 8 c) except for the eastern equatorial Atlantic and Pacific, and the Southern Red Sea and Gulf of Aden, where it is most often a contributor also to the decline phase (Figure 8 d). As the composite amplitude of atmospheric forcing and advective convergence are similar between MHWs onset and decline, the composite event average amplitude for each resembles the positive versus negative patterns described for the onset (except values are much smaller; Figure 7a d). The patterns for the composite event average amplitude of diffusive convergence (Figure 7 g) are instead similar to what seen for the decline phase (Figure 7 h), as, while the duration of the decline phase is on average shorter, the composite amplitude of diffusive divergence (i.e., negative values of diffusive convergence, corresponding to heat loss) is larger for the decline phase (Figure 7 i). The spatial patterns of the composite amplitude of diffusive convergence during MHWs onset and decline are similar to what described for atmospheric forcing (except during the onset phase in proximity of western boundary currents and the Antarctic Circumpolar Current), yet

**Figure 5.** As in Figure 3, now for the average intensity of MHW events. Values in panels a-f have units of  ${}^{\circ}$ C, values in panels g-i are expressed as J/m<sup>2</sup>.

values for diffusive convergence are smaller (Figure 7h, i) and diffusive divergence is most often a contributor globally only to the decline phase, being a leading term at high latitudes (Figure 9e, f). On average, the amplitude of diffusive divergence during the decline phase is larger in regions characterized by strong oceanic fronts, such as western boundary currents and the Antarctic Circumpolar Current (panel i of Figure 7). In these areas, diffusive processes help dissipate warm anomalies, suggesting that turbulent mixing is a key mechanism in reducing excess heat and restoring thermal equilibrium (Oliver et al., 2021; Vogt et al., 2022). While diffusive convergence is generally not a leading term for MHW onset (except near Antarctica) there are events when this happens, indicating that a variety of leading dynamical mechanisms is seen in most regions. Also, diffusive convergence contributes to the onset phase most often in some of the tropics, off the equator (Figure 8e, f).

**Figure 6.** Number (panels a, d), duration (panels b, e) and intensity (panels c, f) of MHW events from 2004 to 2017 detected using ECCO v4r4 ocean heat content between 5-55m (panels a, b, c) and Argo ocean heat content between 15-50dbar (panels d, e, f).

## 4.3 Case studies in the Pacific Ocean

In the following, we discuss more details on the leading dynamical mechanisms of MHWs in three regions in the Pacific ocean that have experienced at least one particularly intense MHW event: NEP, SWP, and TASMAN (introduced in Section 3.2 and shown as boxes in Figure 1). While air-sea exchanges of heat play a key role for the onset and decline of most MHW events, there is diversity in the driving mechanisms of the two phases, both across regions and across events within each of the three regions. In some cases, ocean advective and diffusive processes (convergence and divergence) dominate the heat budget, emerging as leading contributors to MHW onset or decline.

|                    | Terms | NEP (20 events) |         | SWP (38 events) |         | TASMAN (37 events) |         |
|--------------------|-------|-----------------|---------|-----------------|---------|--------------------|---------|
|                    |       | Onset           | Decline | Onset           | Decline | Onset              | Decline |
| "the" leading term | F     | 40-45%          | 25-35%  | 71%             | 29-32%  | 38-49%             | 38-41%  |
|                    | A     | 15-25%          | 10-15%  | 13%             | 18%     | 22%                | 5%      |
|                    | D     | 10%             | 20%     | 5%              | 3%      | 0-3%               | 11-16%  |
| "a" leading term   | F     | 10-25%          | 20-30%  | 11%             | 37-39%  | 24-32%             | 32-35%  |
|                    | A     | 15-25%          | 15%     | 0%              | 13%     | 14-19%             | 5-8%    |
|                    | D     | 5-10%           | 25-35%  | 11%             | 29-32%  | 11-19%             | 32-35%  |

**Table 1.** How often each budget terms is "the" leading term vs "a" leading term (excluding overlap with "the") across NEP, SWP, and TASMAN regions.

**Figure 7.** Average heat budget terms across MHW events with a minimum duration of 5 days, based on ECCOv4r4 upper ocean (5-55 m) heat content: for each event, the average is computed during the whole event (left panels; a, d, g), the onset phase (middle; b, e, h) and the decline phase (right; c, f, i). Panels a-c (top row) show atmospheric forcing, panels d-f (middle row) show advective convergence of heat, panels g-i (third row) diffusive convergence of heat.

220

During the time period analyzed here, 20 MHW events are observed in the NEP region, with five of them related to the Northeast Pacific marine heatwave of 2013-2016, also known as the "Blob" (Events 15-19 in Figure 11a and in Figure S2 in the Supplement). The "Blob" was characterized by an extensive area of anomalously warm sea surface temperatures. It developed due to decreased surface cooling and reduced Ekman transport of colder water from the north, driven by persistent high-pressure systems in the region (Bond et al., 2015; Hartmann, 2015; Di Lorenzo and Mantua, 2016). This event was further sustained by ENSO-related Pacific climate variability through atmospheric teleconnections, which played a critical role in prolonging its duration and amplifying its impacts (Capotondi et al., 2022; Ren et al., 2023; Xu et al., 2022). Consistent with previous studies, Events 15-18 in Figure 11a show the leading contribution of atmospheric forcing to the onset phase, with additional contributions, in some cases substantial (e.g., Events 16-17), from advective or diffusive convergence of heat. Also, advective convergence of heat is the key driver of the onset of MHW Event 19 (Figure 11a), highlighting the importance of multiple

**Figure 8.** Percentage of times each heat budget term contributes to (a, c, e) the MHW onset phase and (b, d, f) the MHW decline phase, based on ECCOv4r4 upper ocean (5-55 m) heat content. A budget term contributes to the onset phase if it is positive, to the decline phase if it is negative. Percentages are shown for (a, b) atmospheric forcing, (c, d) advective convergence of heat and (e, f) diffusive convergence of heat.

mechanisms working together to sustain MHW intensity and prolong its duration. The "Blob" also showed how changes in atmosphere-ocean interactions in response to initial SST anomalies (including a reduction of low-cloud cover enhancing insolation), may further sustain and intensify these anomalies, increasing their persistence (Myers et al., 2018; Schmeisser et al., 2019).

Overall, in the NEP region, atmospheric forcing is a key driver of MHW onsets, leading the onset phase (alone or together with other processes) in about 50-70% of the cases (see tags that include "F" in Figure 11a; also, see the sum of the percentages for "F" in the NEP onset column of Table 1) and being "the" leading process in about 40-45% of the events (see tags with only "F" in Figure 11a; e.g., Event 3). Heat transport by ocean currents is another key process during MHW onset, leading this phase (alone or together with other processes) in 30-50% of the cases and being "the" leading mechanism in four events (Events 10, 19, 20, 21 in Figure 11a). While diffusive convergence of heat is "the" leading mechanism for MHW onset in the NEP in the time period of interest in just two instances (Events 5 and 13 in Figure 11a), it contributes together with air-sea exchanges

**Figure 9.** Same as Figure 8, now for the percentage of times each term is a leading contributor to the MHW (a, c, e) onset and (b, d, f) decline phase. As an example, we include in the count for panel (a) both a case where atmospheric forcing is the only contributor to the onset phase and a case where atmospheric forcing is a leading contributor together with advective and/or diffusive convergence of heat.

and/or advective convergence in 5-10% of the cases, i.e., Events 11, 16 in Figure 11a. Also, diffusive divergence of heat leads MHW decline (alone or together with other processes) in 45-55% of the events, e.g., see tags including "D" in Figure 11b. In two of these events, diffusive divergence of heat leads the decline phase together with advective divergence of heat ("DA" and "AD" tags); in 4 events, diffusive divergence is "a" leading mechanism together with atmospheric forcing (see "DF" or "FD" tags in Figure 11b). Overall, atmospheric forcing leads (alone or together with other processes) MHW decline in the region for about 45-65% of the events (see tags including "F" in Figure 11b, except if the tag has three letters and "F" is the last one) and is "the" leading process in 6, e.g., Events 7, 13, 16, 18, 19, 21. Advective divergence of heat is "the" leading mechanism in 3 events instead (Events 5, 11 and 14) and is "a" leading process in 15% of the cases (together with forcing and/or diffusive divergence of heat). This is consistent with a diversity of leading dynamical mechanisms for MHWs in the NEP, both during onset and decline (Table 1).

240

245

Diversity of leading mechanisms also emerges across the 38 MHW events in the SWP region (Table 1). In this region, extreme extra-tropical MHWs appear to be associated with different phases of El Nino Southern Oscillation (Dutheil et al.,

**Figure 10.** Same as Figure 8, now for the percentage of times each term is the smallest contributor to the MHW (a, c, e) onset and (b, d, f) decline phase (however similar to other terms it may be).

2024; Gregory et al., 2024a). Indeed, MHW conditions were detected during the 2010/11 La Nina event (Boening et al., 2012), corresponding to event 30 in Figure 12 and Figure S4c, d in the Supplement. The onset phase of this event was primarily driven by atmospheric forcing, consistent with about 71% of SWP MHWs (e.g., see Events 2, 12, 29, 30 with tags "F" in Figure 12a). This agrees with previous studies indicating that air-sea exchanges often dominate MHW development in the region (Sen Gupta et al., 2020). We also note that in an additional 11% of events the atmospheric forcing is a leading term during the onset together with advection and/or diffusion (e.g., Events 5, 20, 37 in Figure 12a). Heat transport by ocean currents, i.e., advective convergence of heat, is "the" leading process of SWP MHW onset in 13% of events (Events 4, 6, 10, 11, 14 in Figure 12a), while diffusive convergence of heat is "the" leading mechanism in only 2 cases (e.g., Events 18, 27 in Figure 12a) and contributes alongside forcing and/or advection in about 11% of cases (e.g., Events 20, 37 in Figure 12a). As for the onset, atmospheric forcing leads MHW decline (alone or together with other processes) in most cases (about 66-71% of the events, e.g., Event 30 in Figure 12b and Figure S4c in the Supplement). Atmospheric forcing leads MHW decline together with diffusive divergence of heat in about 26% of the events (e.g., Event 11 in Figure 12b and in Figure S4b in the Supplement) and with advective convergence in about 11% of the events (e.g., Event 33, 34 in Figure 12b). Also, advective divergence of heat is

**Figure 11.** MHW events in the Northeast Pacific (NEP, region 1 in Figure 1) based on ECCOv4r4 upper ocean (5-55 m) heat content: ratio of the total contribution for each budget term during each phase (onset/decline), as well as the tendency (scaled too by the total contribution). The top panel is for the onset phase (a), the bottom one for the decline phase (b). Each stacked bar represents the relative contribution of each term during that phase, with the total contribution (i.e., the sum of the terms that contribute to that phase) normalized to 1. The black outline over each bar indicates the total temperature tendency during that phase (scaled by the total contribution to that phase), showing agreement with the sum of the individual terms and confirming budget closure. The x-axis labels denote the start date of each MHW event, followed by values in parentheses indicating the event intensity (as average temperature (degC) in the layer used for the OHC estimate), onset/decline duration (in days), and total duration. The letter codes above each bar indicate which term(s) dominated the total temperature tendency during that phase (A = advection, D = diffusion, F = forcing), e.g., "F" is used for cases when the forcing provides the leading contribution, "FA" is used when both forcing and advective convergence provide a leading contribution, "AFD" characterizes cases where advective convergence is larger than forcing and diffusive convergence and forcing is larger than diffusive convergence, yet the difference does not meet the 30% criteria as in the previous two cases. Finally, " " corresponds to cases where all terms contribute comparably.

**Figure 12.** As in Figure 11, now for the Southwest Pacific region (SWP, region 2 in Figure 1).

"the" leading term for MHW decline in 18% of the events (e.g., Event 35 in Figure 12b and in Figure S4f in the Supplement), and contributes alongside forcing and/or diffusion in another 13% (e.g., Events 15 in Figure 12b). Diffusive divergence of heat is "the" leading mechanism in only one MHW decline case (e.g., Event 25 in Figure 12b), yet it is "a" leading term alongside other processes in 29-32% of the cases (e.g., Events 5, 11 in Figure 12b).

Finally, 37 MHW events are found in the TASMAN region, where again we see a diversity of driving mechanisms (Table 1). Five of the events (Events 30 to 34 in Figure 13 and in Figure S5k-r in the Supplement) are associated with the intense 2015-2016 Tasman Sea marine heatwave described in Oliver et al. (2017). This unprecedented warming event, characterized by sustained heat anomalies, was attributed to anomalous convergence of heat linked to the intensification of the southward flowing East Australian Current (Kajtar et al., 2022), making it the longest and most intense MHW on record in the region (Oliver et al., 2017). Our results confirm that advective convergence of heat is a primary driver during the onset phase of Events 30, 31, and 34 (in Figure 13a and Figure S5k, o in the Supplement). More generally, advective convergence of heat is "a" leading process (alongside other mechanisms) for 14-19% of onset cases and "the" leading process in 22% of events (e.g., Events 1, 18, 34 in Figure 13a). Overall, as for the NEP and SWP, atmospheric forcing provides a key contribution to

MHW onset in the TASMAN region in the majority of events (≈62-81%), including about 38-49% of events when it is "the" primary driver of the MHW onset (see tags with only "F" in Figure 13a; e.g., Events 3, 4, 11, 35 in Figure 13a and in Figure S5 in the Supplement). This anomalous atmospheric forcing may originate from a variety of remote influences, associated with various modes of variability (e.g., ENSO, Indian Ocean Dipole, Southern Annual Mode, etc.), as discussed in Gregory et al. (2023, 2024b), who used this information to inform MHW event predictability. Diffusive convergence of heat, while not being the only leading mechanism in any of the cases, is "a" leading process in about 11-19% of the onset phases. In the TASMAN region, atmospheric forcing alone leads the decline phase in the region in around 38-41% of events (e.g., Event 31 in Figure 13b and in Figure S51 in the Supplement). In 2 events, air-sea exchanges of heat act together with advective divergence of heat as the main drivers of the decline phase, e.g., Event 9 (Figure 13b). Forcing together with diffusive divergence of heat leads the MHW decline in about 30% of the events instead (see tags with "FD" or "DF" in Figure 13, e.g., Event 12 in Figure S5d in the Supplement). Additional decline- phase cases include two primarily driven by advective divergence of heat (e.g., Event 3 in Figure 13b) and four led by diffusive divergence of heat (e.g., Event 25 in Figure 13 and in Figure S5h in the Supplement). Overall, advective and diffusive divergence of heat are a leading term for MHW decline in 5-8% versus 32-35% of events, respectively.

#### 5 Conclusions

In this study, we have conducted a comprehensive analysis of marine heatwaves using various observational and reanalysis datasets to understand their characteristics and the leading dynamical processes that drive their formation and decline. We validate the ECCO ocean reanalysis data against observational data from OISST and Argo floats: ECCO is overall consistent with observations and critical for precisely assessing the processes responsible for MHW onset and decline, although it tends to underestimate the number and intensity of MHW events, especially at the daily timescale, and overestimate their duration, as also found for other models (see Capotondi et al. (2024), for a review).

We explore the heat budget terms contributing to the time evolution of MHWs, focusing on atmospheric forcing, advective convergence, and diffusive convergence of heat. Our results indicate that atmospheric forcing is the dominant contributor to both the onset and decline phases of MHWs across most oceanic regions, with the exception of equatorial regions and of some locations in the extra-tropics and in the Southern Ocean, where ocean dynamics plays a key role. Advective convergence and divergence of heat are leading dynamical mechanisms for MHWs near the Equator, especially in the Pacific, where extreme warming is closely related to ENSO variability, in western boundary currents, and (for the onset phase) along the Antarctic Circumpolar Current. Diffusive convergence and divergence of heat lead most often MHW onset and decline phases, respectively, in many regions of the Southern Ocean, except where the advective term is "the" leading dynamical mechanism.

Our analysis of the Northeast Pacific, Southwest Pacific, and Tasman Sea regions highlights the complex interplay of oceanic and atmospheric processes driving MHWs, underscoring the need for regional-scale studies to capture the details of local dynamics for different events. While atmospheric forcing plays a key role in MHW onset and decline across the three regions, in most cases, diverse combinations of processes are at play across regions and across events in the same region. Also, in

Figure 13. As in Figure 11 and 12, now for the Tasman Sea region (TASMAN, region 3 in Figure 1).

315

the NEP and TASMAN region, advective convergence of heat is a leading term during MHW onset more often than diffusive convergence, while the opposite is true for the decline phase. In the SWP region, the two processes are leading mechanisms nearly as often, both during onset and decline, instead.

Our findings highlight the importance of using daily data to study leading dynamical processes during MHWs onset and decline and emphasize the importance to compare ocean state estimates and reanalyses with a suite of observations to better understand strengths and limitations of these products. Understanding MHWs leading dynamical processes and their regional variations is essential for predicting MHWs and assessing their potential impacts on marine ecosystems and global climate patterns. Future research should aim to further refine these analyses by incorporating more regional studies of MHWs in the present and future climate and exploring the interaction between physical processes and biological responses to MHWs.

Data availability. NOAA OI SST V2 High Resolution Dataset data provided by the NOAA PSL, Boulder, Colorado, USA, from their website at https://psl.noaa.gov.

ECCO Version 4 Release 4 Dataset ECCO Consortium, Fukumori, I., Wang, O., Fenty, I., Forget, G., Heimbach, P., & Ponte, R. M. (2022-04-12). ECCO Central Estimate (Version 4 Release 4). Retrieved from https://ecco.jpl.nasa.gov/drive/files/Version4/Release4.

ECCO Version 4 Release 4 Synopsis ECCO Consortium, Fukumori, I., Wang, O., Fenty, I., Forget, G., Heimbach, P., & Ponte, R. M. (2021, February 10). Synopsis of the ECCO Central Production Global Ocean and Sea-Ice State Estimate (Version 4 Release 4). https://doi.org/10.5281/zenodo.4533349.

ECCO Version 4 Description Forget, G., J.-M. Campin, P. Heimbach, C. N. Hill, R. M. Ponte, and C. Wunsch, 2015: ECCO version 4: An integrated framework for non-linear inverse modeling and global ocean state estimation. Geoscientific Model Development, 8. https://www.geosci-model-dev.net/8/3071/2015/.

Argo data were collected and made freely available by the International Argo Program and the national programs that contribute to it. (https://argo.ucsd.edu, https://www.ocean-ops.org). The Argo Program is part of the Global Ocean Observing System.

Author contributions. JS, DG, AC contributed to the conceptualization and design of this study. JS conducted the analysis and wrote the manuscript, with contributions from DG, AC, and MK. TS, DG, MK contributed to data curation. The final manuscript underwent a thorough review and editing process, led by JS, DG, AC, MK and TS, ensuring its quality and accuracy.

Competing interests. The author has declared that none of the authors has any competing interests.

325

Acknowledgements. JS, DG, and AC have been supported by NASA award #80NSSC21K0556. JS and DG have also been supported by NOAA award #NA21OAR4310261, and DG by KAUST CRG Grant ORA-2021-CRG10-4649.2. This work used Bridges-2 at Pittsburgh Supercomputing Center through allocation #EES230075 from the Advanced Cyberinfrastructure Coordination Ecosystem: Services & Support (ACCESS) program, which is supported by National Science Foundation grants #2138259, #2138286, #2138307, #2137603, and #2138296.

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
