# Peer review of "Leading Dynamical Processes of Global Marine Heatwaves in an Ocean State Estimate"

_EGUsphere, 2025_

## Author Comment (AC1)

We thank the reviewer for their thoughtful comments. In the responses below, the reviewer comments are in bold and italic, and our responses are in normal font.

**REVIEW 1**

*The manuscript presents an effort to characterize the physical drivers of marine heatwaves (MHWs) using the ECCO state estimate framework. The authors should be commended for undertaking the challenge of quantifying the upper ocean heat budget globally and attributing MHWs to specific dynamical processes. However, there are several important concerns that limit confidence in the presented results. First, there is a notable mismatch between the ECCO product and observations, raising questions about the representativeness and reliability of the diagnosed drivers. Second, there are questions regarding the heat budget closure and identification of dominant processes, complicating the interpretation of the contributions from different terms. These issues, along with several concerns related to the interpretation and presentation of results, warrant further consideration and clarification.*

Major concerns:

1. *Comparisons between ECCO and OISST reveal clear mismatches in the number, duration, and intensity of MHWs globally (Figures 3–5). This raises the question: how representative are MHWs in ECCO of those in the real ocean? The title of the manuscript is ambitious, yet the results are evidently dependent on the fidelity of the ECCO product.*

We acknowledge the differences between ECCO and OISST in the number, duration and intensity of MHWs, as shown in Figures 3–5. Some of these differences may be related to ECCO's resolution: Pilo et al. (2019) show that while model configurations with resolutions ranging from 1° to 1/10° degree, can all qualitatively represent broad-scale global patterns of MHWs, modeled MHWs are weaker, longer-lasting, and less frequent than in observations, especially for models with lower resolution.

This is due in part to smoother SST time series and longer autocorrelation times (Cooper et al., 2017) in models compared to observations, which can suppress some of the short-lived variability and artificially extend the duration of events. High-resolution, eddy-permitting models perform generally better, particularly in dynamic regions like western boundary currents, but still exhibit biases (Pilo et al., 2019).

However, the differences we see between ECCO and OISST do not diminish the value of our analysis. Despite its limitations, which are common to most modeling products, ECCO offers unique advantages for MHW research. Our study complements previous work based on free-running climate models (of comparable resolution to ECCO) by leveraging an ocean state estimate that is constrained by observations and is dynamically consistent. ECCO's ability to close the heat budget exactly makes it uniquely suited for budget analyses. This provides a critical perspective on MHW dynamics that observational products alone cannot offer, thereby justifying the broader scope of the manuscript. Also, ECCO compares well with observational estimates of MHW based on upper ocean heat content (Figure 6a-c based on ECCO, versus Figure 6d-f based on Argo observations).

In the revised manuscript, we introduced a new title:

"**Leading Dynamical Processes of Global Marine Heatwaves in an Ocean State Estimate**"

This text (bold font embedded in the text from the original submission for context) was also included in Section 4.1:

ECCO provides an overall good representation of **the spatial patterns** of both the long term linear trend in upper ocean temperature (e.g., Figure 1) and the 90th percentile anomalies (Figure S1 in the Supplement); also, spatial patterns of MHWs frequency, average duration, and average intensity in ECCO are consistent with observations (Figures 3-6). Yet, a smaller number of near surface MHW events shorter than a month (i. e. duration between 5 and 29 days) is seen in ECCO compared to observations (Figure 3a d), with only some of these events showing a signature in upper (5-55m) ocean heat content (Figure 3 g).

**Some of the differences between ECCO and observations may be related to ECCO's resolution: Pilo et al. (2019) show that while model configurations with resolutions ranging from 1° to 1/10° degree, can all qualitatively represent broad-scale global patterns of MHWs, modeled MHWs tend to be weaker, longer-lasting, and less frequent than in observations, especially for models with lower resolution. High-resolution, eddy-permitting models perform generally better in representing MHW characteristics, particularly in dynamic regions like western boundary currents, but still exhibit biases (Pilo et al., 2019).**

**Discrepancies between models and observations are due in part to smoother SST time series and longer autocorrelation times (Cooper et al., 2017) in models (compared to observations), which can suppress short-lived variability and artificially extend the duration of events**, and it may reflect limitations in how the thermal memory of the ocean is represented in ECCO. Factors such as the deepening of the mixed layer and a reduction in heat loss rates can extend the persistence of SST anomalies, phenomena that are challenging to fully capture in models (Lee et al., 2024). Also, short lived MHWs are most common in regions of strong temperature fronts and eddy variability (e.g., Western Boundary Currents), where the variance is high due to relatively rapid transport variations and large mesoscale eddy activity compared with other areas (Holbrook et al., 2019; Bian et al., 2024), which are challenging features to represent in coarser resolution models like ECCO. The number of MHW events lasting 30 days or longer is, instead, larger in ECCO than in observations (Figure 3b, e), with a similar pattern between MHWs based on ECCO near surface temperature and upper ocean heat content (Figure 3e, h), indicating that many of these events have a signature both at the surface and in the subsurface. More events are detected in OISST (compared to ECCO) again when analyzing monthly fields (Figure 3e, f), due to the effect of shorter (5-29 days) MHW events on the monthly averages. This effect also explains the difference between panels b, e, h (for 30+ days events) and c, f, i (for monthly events) in Figure 3. While long lasting MHW events are not as frequent as short-lived events both in ECCO and in observations (center versus left column in Figure 3), the spatial distribution of the number of events is more similar (between long- versus short- lived MHWs) to the observations. Consistent with the discussion of the number of MHWs in ECCO versus observations, the average duration of MHW events is overall longer in ECCO for all event lengths and especially short-lived MHWs (Figure 4), indicating limitations in how ECCO captures the thermal memory of the ocean. Also, while global

patterns of the average MHW intensity are well represented in ECCO, the intensity is underestimated compared to observations **(consistent with Pilo et al. 2019)**, both for near surface temperature and upper ocean heat content (Figure 5 and 6). We note that MHWs characteristics in ECCO agree overall better with observations when considering monthly upper-ocean heat content (Figure 6) than near surface temperature (Figures 3-5), as (1) OHC is an integrated quantity, and (2) there are limitations in how both ECCO and gridded fields from sparse Argo observations represent the complexity of the real ocean. (While Argo provides unprecedented coverage of the subsurface ocean globally, a product based on Argo observations incorporates a smaller number of measurements compared to OISST, hence gridded Argo fields may not capture some of the ocean variability.)

*Capotondi, Antonietta, et al. "A global overview of marine heatwaves in a changing climate." Communications Earth & Environment 5.1 (2024): 701.*

*Cooper, Fenwick C. "Optimisation of an idealised primitive equation ocean model using stochastic parameterization." Ocean Modelling 113 (2017): 187-200.*

*Pilo, Gabriela S., et al. "Sensitivity of marine heatwave metrics to ocean model resolution." Geophysical Research Letters 46.24 (2019): 14604-14612.*

2. *The calculation of heat budget terms, which is key to the manuscript, is not clearly described. Offline calculations of the heat budget often fail to close due to the use of temporally averaged velocity and temperature fields, which do not account for nonlinear covariance terms like $\nabla \cdot (\vartheta' T')$. Depending on the region, these covariance terms can be significant. The authors do not report how well the budget closes or the magnitude of the residual term, which raises questions about the accuracy of the diagnosed contributions from different processes.*

We appreciate the reviewer's comment and the opportunity to clarify this important point. We chose the ECCOv4r4 product as it is a dynamically consistent ocean state estimate and, while constrained by observations, allows for heat budget closure (Forget et al., 2015).

However, although not visible (as orders of magnitude smaller), the residual term is displayed in Figures 11-13 (shown below, in response to another comment). Also, we show here (at the end of our response and as additional figure in the supplemental) the residual term corresponding to Figure 7 (which is orders of magnitude smaller than budget terms).

Finally, we included this sentence in Section 3.2 (ECCO ocean heat budget):

"**We note that ECCO ocean heat budget closes practically exactly, and residuals are orders of magnitude smaller than the other terms.**"

[Figure]

Maps representing the heat budget residual for the events average (left), the onset phase (middle) and the decline phase (right).

Forget, G., J.-M. Campin, P. Heimbach, C. N. Hill, R. M. Ponte, and C. Wunsch, 2015: ECCO version 4: an integrated framework for non-linear inverse modeling and global ocean state estimation. Geoscientific Model Development, 8, 3071-3104, http://dx.doi.org/10.5194/gmd-8-3071-2015, http://www.geosci-model-dev.net/8/3071/2015/

3. ***The partitioning of contributions into advective and diffusive terms is inherently tied to the model's resolution. Coarse-resolution products may misattribute unresolved advective processes to subgrid-scale diffusion. Without acknowledging this limitation, the attribution results risk being potentially misleading.***

We thank the reviewer for raising this important point. We agree that the attribution of heat budget terms, particularly the separation of advection and diffusion, is influenced by model resolution. Coarse-resolution models may indeed misattribute unresolved advective processes to subgrid-scale diffusion. We have clarified this limitation in the manuscript (see text at the end of this response). However, ECCO's resolution is comparable to that of free running models used in other studies of marine heat waves, e.g., Deser et al. 2024 (CESM large ensemble, between 1° and 1.5°) and Vogt et al. 2022 (MOM4p1, with a nominal 1° resolution increasing (in longitude) to 1/3° near the equator). While higher-resolution, eddy-permitting (~0.25°) or eddy-rich (~0.1°) models improve the realism of MHW characteristics (especially in regions like western boundary currents), Pilo et al. (2019) show that even coarse resolution models can qualitatively capture broad-scale MHW patterns in less active regions. For global, process-based analyses such as ours, ECCO offers a balance between resolution, dynamical fidelity (thanks to being constrained by observations), and global coverage. Moreover, ECCO enables full closure of the heat budget, which is not possible with many other data assimilating products, e.g., GLORYS. This makes ECCO particularly well-suited for process-based MHW analysis, even within the known constraints of resolution. Finally, as described in Pilo et al. (2019), while all models (including those with high resolution) exhibit systematic biases in MHW frequency, intensity, and duration, 1x1 degree models can represent broad-scale global patterns of MHWs. ECCO's strengths, particularly in dynamically consistent heat budget closure and observational constraints, make it a valuable tool for understanding processes underlying MHWs, even if some subgrid-scale processes remain parameterized.

We included the following text in the last paragraph of the introduction:

"**This dynamically consistent ocean state estimate incorporates a wide range of oceanic and atmospheric observations (Forget et al., 2015) and provides daily temperature fields as well as all heat budget terms computed at each model time step, ensuring closure of the heat budget. We note that the attribution of heat budget terms, particularly the separation of advective and diffusive contributions, is inherently tied to model resolution, and coarse-resolution products may misattribute unresolved advective processes to subgrid-scale diffusion. Yet, while eddy-permitting (~0.25°) or eddy-rich (~0.1°) models improve MHW realism in highly dynamic regions (e.g., western boundary currents), even coarser-resolution models can qualitatively capture large-scale MHW patterns in less active regions (Pilo et al., 2019). While ECCO's 1x1 degree resolution is a limitation, results from (observations constrained) ECCO's results provide an excellent complement to results from free-running models (of comparable resolution)**

described in recent MHW studies, such as work using the CESM large ensemble (1°–1.5°; Deser et al., 2024) and an analysis based on the GFDL ESM2M coupled Earth System Model (with a nominal 1° resolution increasing (in longitude) to 1/3° near the equator; Vogt et al., 2022, describing MHW dynamics and local drivers of MHW onset and decline phases in different seasons, over a 500-year period). ECCO provides an optimal balance between resolution, dynamical fidelity (thanks to being constrained by observations), and spatial coverage for global, process-based MHW analysis. Also, compared to other data assimilating products, ECCO enables heat budget closure—an essential advantage for our process-focused analysis."

*Deser, Clara, et al. "Future changes in the intensity and duration of marine heat and cold waves: insights from coupled model initial-condition large ensembles." Journal of Climate 37.6 (2024): 1877-1902.*

*Forget, G. A. E. L., et al. "ECCO version 4: An integrated framework for non-linear inverse modeling and global ocean state estimation." Geoscientific Model Development 8.10 (2015): 3071-3104.*

*Pilo, Gabriela S., et al. "Sensitivity of marine heatwave metrics to ocean model resolution." Geophysical Research Letters 46.24 (2019): 14604-14612.*

*Vogt, Linus, et al. "Local drivers of marine heatwaves: a global analysis with an earth system model." Frontiers in climate 4 (2022): 847995.*

4. *The identification of leading terms is based on predefined thresholds (30%, 16%) that are not rationalized. The conclusions drawn in Sections 4.2 and 4.3 are thus sensitive to these subjective choices, which undermines their robustness.*

We appreciate the reviewer's comment regarding the use of predefined thresholds. In the revised manuscript, we implemented a simplified criterion for the identification of leading terms (described later in this response): while the selected threshold is somewhat arbitrary, it helps summarize our results. Also, we clarify that our results are robust to $\mp5\%$ changes in the selected threshold (see figures at the end of this response).

While Figures 9 (shown at the end of this response), 11-13 (shown later, in response to another comment) were updated to reflect the new (simplified) method, results are consistent with the previous version of the plots. We also now show a range for the values in Table 1, to indicate how the percentages change with a threshold value of 25% and 35% instead of 30% (see table at the end of this message). As discussed for the overall classification of leading terms above, the purpose of the table is to summarize which processes are most often leading terms, and the focus is not the exact percentage value reported.

In the revised manuscript we included this text (in the methods section) to introduce the simplified criteria for the identification of leading terms:

"**Towards summarizing our global and regional findings for the leading dynamical processes driving MHWs, we introduce the terms "the" leading term and "a" leading term, defined as follows. For each phase, we identify the budget terms that contribute to it (among forcing, advective convergence, diffusive convergence), then we sort them by the magnitude of the contribution. A term provides "the" leading contribution if it exceeds the next largest term by at least 30%. A term also provides the leading contribution if it is the only process contributing**

to the phase of interest. If the largest two contributors are both greater than the third by at least 30%, but neither is larger than the other by 30%, then each of the two terms provides "a" leading contribution. The same happens if these two terms are the only contributors and neither is larger than the other by 30%. If none of the terms provides a contribution that is 30% larger than other contributions, all three terms contribute comparably. We note that while the 30% threshold is arbitrary, it serves the purpose of summarizing our results, and visually identifying which processes are most often leading terms. Our findings are robust to $\mp 5\%$ change in the (30%) threshold percentage used (not shown)."

Updated Table 1:

| | Terms | NEP (20 events) | | SWP (38 events) | | TASMAN (37 events) | |
|---|---|---|---|---|---|---|---|
| | | Onset | Decline | Onset | Decline | Onset | Decline |
| | F | 40-45% | 25-35% | 71% | 29-32% | 38-49% | 38-41% |
| "the" leading term | A | 15-25% | 10-15% | 13% | 18% | 22% | 5% |
| | D | 10% | 20% | 5% | 3% | 0-3% | 11-16% |
| | F | 10-25% | 20-30% | 11% | 37-39% | 24-32% | 32-35% |
| "a" leading term | A | 15-25% | 15% | 0% | 13% | 14-19% | 5-8% |
| | D | 5-10% | 25-35% | 11% | 29-32% | 11-19% | 32-35% |

How often each budget term is "the" leading term vs "a" leading term (excluding overlap with "the") across NEP, SWP, and TASMAN regions. A range of percentage values is shown for each case, indicating how percentages change with a threshold value of 25% and 35% instead of 30%, in the definition of "the" vs "a" leading term.

Updated Figure 9:

[Figure]

Figure 9. Same as Figure 8, now for the percentage of times each term is a leading contributor to the MHW (a, c, e) onset and (b, d, f) decline phase. As an example, we include in the count for panel (a) both a case where atmospheric forcing is the only contributor to the onset phase and a case where atmospheric forcing is a leading contributor together with advective and/or diffusive convergence of heat.

Versions of Figure 9 using 25% and 35% (instead of 30%) in the criteria to define "the" versus "a" leading term:

25%

35%

[Figure]

**Other comments:**

- *Page 7, The claim that "ECCO provides an overall good representation..." is somewhat overstated. ECCO underestimates the magnitude of the trend in the Northwest Pacific, Northwest Atlantic, and Southwest Atlantic. The spatial patterns in MHW metrics also show clear mismatches in Figures 3–6.*

Thank you for your comment. Regarding the trend, we included a revised statement:

"ECCO provides an overall good representation of **the spatial patterns** of both the long term linear trend in upper ocean temperature (e.g., Figure 1) and the 90th percentile anomalies (Figure S1 in the Supplement); ..."

Regarding MHW characteristics we revised the discussion as indicated earlier in response to other comments.

- *Page 9, The statement that "intensity is slightly underestimated..." is problematic. ECCO upper ocean heat content and OISST surface temperatures are not directly comparable.*

Thank you for the opportunity to clarify. We compared the ECCO upper OHC with OHC from Argo (Figure 6), while we compared ECCO near surface temperature with OISST (Figure 5).

Revised sentence:

"**Also, while global patterns of the average MHW intensity are well represented in ECCO, the intensity is underestimated compared to observations (consistent with Pilo et al. 2019), both for near surface temperature and upper ocean heat content (Figure 5 and 6).**"

- *Page 10, "In these regions" should be "in other regions"?*

We edited the sentence to clarify the continued discussion about the advective convergence:

"**On average, advective convergence contributes to both the onset and decline phase only near the equator (panel e, f in Figure 7), in the subtropics and western boundary currents (consistent with previous studies, (e.g., Li et al., 2022; Zhang et al., 2023), in the Southern Red Sea and Gulf of Aden (Nadimpalli et al., 2025), and along the Antarctic Circumpolar Current (Figure 7e, f). In these regions, the advective convergence is most often a contributor only to the onset phase (Figure 8 c) except for the eastern equatorial Atlantic and Pacific, and the Southern Red Sea and Gulf of Aden, where it is most often a contributor also to the decline phase (Figure 8 d).**"

- *Page 11, "Figure 8a d" should be "Figure 7a d".*

Thank you for noticing the typo. The sentence was removed in response to the next comment.

- *Page 11, "consistent with the onset phase being on average longer than the decline phase (Figure 4)." Why longer duration means overall pattern being similar to that of*

> *onset, especially the counter-argument is made later in the same paragraph? Besides, Figure 4 doesn't show onset vs decline.*

Thank you for the comment; we agree this sentence can be confusing, hence we have removed it from the revised manuscript.

- *Page 14, "How often each budget terms is "the" leading term vs "a" leading term (excluding overlap with "the") across NEP, SWP, and TASMAN regions". The definition of "the" and "a" leading term needs to be clearly explained.*

Thank you for the comment. As described above, we have simplified our approach for how leading terms are identified and introduced text to clarify the definition.

- *Pages 15-18, Figures 11-13: 1. the percentage contribution should be based on the total temperature change, i.e., , where is the temperature change during onset or decline, so the contribution from one term can exceed 100%, if there is one negative term or more. 2. the numbers are only meaningful if the heat budget is closed, that is, equals . Otherwise, if there are residuals in the budget, these numbers are less meaningful.*

We thank the reviewer for the constructive suggestions regarding the calculation and presentation of percentage contributions in Figures 11–13. While we agree that the reviewer's suggestion (to calculate percentage contributions relative to the total temperature tendency) is helpful to show how each term compares with the tendency, the resulting figure (we include one example below) reduces the visibility of the main information we would like the figure to show (due to how the relative amplitudes of the budget terms change across events).

The key goal for these figures is to show, at a glance, which terms are most often the largest contributors to the phase of interest across events: scaling the terms by the total contribution to the phase of interest allows for that.

Here is an example for how the figure would look like if percentage contributions were calculated relative to the total temperature tendency:

[Figure]

The reviewer's comment inspired us to include (in our updated Figures 11-13, shown below) information on (1) how the tendency (black horizontal outline in the updated plots below) compares to other terms and (2) the overall intensity of the phase of interest: we now show the tendency (as a bar, scaled by the total contribution to the phase of interest) and the intensity (in the x-tick labels, as average temperature in the ocean layer used for the OHC estimate). We note that, however not visible (as orders of magnitude smaller than budget terms), the residual is also shown.

Here are updated Figures 11-13, included in the revised version of the manuscript:

NEP

[Figure]

The figure above (and similarly the ones that follow, for other regions) shows the ratio of the total contribution for each budget term during each phase (onset/decline), as well as the tendency (scaled too by the total contribution). In each figure, the top panel is for the onset phase, the bottom one for the decline phase. Each stacked bar represents the relative contribution of each term during that phase, with the total contribution (i.e., the sum of the terms that contribute to that phase) normalized to 1 . The black outline over each bar indicates the total temperature tendency during that phase (scaled by the total contribution to that phase), showing agreement with the sum of the individual terms and confirming budget closure. The x-axis labels denote the start date of each MHW event, followed by values in parentheses indicating the event intensity (as average temperature (degC) in the layer used for the OHC estimate), onset/decline duration (in days), and total duration. The letter codes above each bar indicate which term(s) dominated the total temperature tendency during that phase (A = advection, D = diffusion, F = forcing), e.g., if the forcing provides the leading contribution ("F"), if both forcing and advective convergence provide a leading contribution ("FA"), if the advective convergence is 30% larger than the diffusive convergence, yet it is not 30% larger than the forcing ("AFD"), if all terms contribute comparably ("~").

SWP

[Figure]

TASMAN

[Figure]

- *Pages 15-18: all the numbers depend on how the percentage of the heat budget terms are calculated, and the threshold used for ranking contributors.*

As discussed in response to an earlier comment, our results are robust to changes around the selected percentage value.

In addition to the updated Table 1 and new text described above, we included this text in the method's section of the revised manuscript:

[revised manuscript text omitted]

---

## Author Comment (AC3)

**REVIEW 3**

We thank the reviewer for their thoughtful comments. In the responses below, the reviewer comments are in bold and italic, and our responses are in normal font.

*This study presents a comprehensive analysis of the leading dynamical processes of global MHWs during the onset and decline periods, based on ECCO and OISST datasets. While the paper is generally clear and the results could contribute to our understanding of MHW evolution, significant shortcomings preclude recommending it for publication in its current form.*

1. *Firstly, the authors used daily and monthly datasets to calculate MHWs after removing linear trends. The spatial patterns of the MHW metrics differ significantly between the OISST and ECCO datasets (see Figures 3 and 4). This suggests that ECCO does not capture MHW characteristics in the same way as observations. Therefore, I suggest that the authors use more ocean reanalysis datasets, such as GLORYS or BRAN.*

Thanks for your comment. We acknowledge the differences between ECCO and OISST in MHW metrics, as shown in Figures 3–4. Some of these differences may be related to ECCO's resolution: Pilo et al. (2019) show that while model configurations with resolutions ranging from 1° to 1/10° degree, can all qualitatively represent broad-scale global patterns of MHWs, modeled MHWs are weaker, longer-lasting, and less frequent than in observations, especially for models with lower resolution.

This is due in part to smoother SST time series and longer autocorrelation times (Cooper et al., 2017) in models compared to observations, which can suppress some of the short-lived variability and artificially extend the duration of events. High-resolution, eddy-permitting models perform generally better, particularly in dynamic regions like western boundary currents, but still exhibit biases (Pilo et al., 2019; please also refer to Fig. R1 here below). For example, Fig. R1 here below shows that while the higher-resolution ocean reanalysis GLORYS, one of the products suggested by the reviewer, does have a better comparison with OISST for MHW duration relative to ECCO (bottom row in Fig. R1), but exhibits an overall large positive bias in MHW frequency (top row in Fig. R1), a metric that is better captured by ECCO.

[Figure]

Fig. R1. MHW metrics (see labels on the left) compared across different products (see labels on the top). GLORYS panels are from Guo et al. 2024.

For these reasons, the differences we see between ECCO and OISST do not seem to diminish the value of our analysis. Despite its limitations, which are common to most modeling products, ECCO offers unique advantages for MHW research. In particular, ECCO's ability to close the heat budget exactly (Forget et al., 2015) makes it uniquely suited for budget analyses. This provides a critical perspective on MHW dynamics that observational products alone cannot offer, thereby justifying the broader scope of the manuscript. Also, ECCO compares well with observational estimates of MHW based on upper ocean heat content (Figure 6a-c based on ECCO, versus Figure 6d-f based on Argo observations).

While higher resolution data assimilating systems are available (e.g., GLORYS, BRAN), they are non-conservative and do not allow for heat budget closure, which is key to our study.

This text (bold font embedded in the text from the original submission for context) was included in Section 4.1 of the revised manuscript:

ECCO provides an overall good representation of **the spatial patterns** of both the long term linear trend in upper ocean temperature (e.g., Figure 1) and the 90th percentile anomalies (Figure S1 in the Supplement); also, spatial patterns of MHWs frequency, average duration, and average intensity in ECCO are consistent with observations (Figures 3-6). Yet, a smaller number of near surface MHW events shorter than a month (i. e. duration between 5 and 29 days) is seen in ECCO compared to observations (Figure 3a d), with only some of these events showing a signature in upper (5-55m) ocean heat content (Figure 3 g).

Some of the differences between ECCO and observations may be related to ECCO's resolution: Pilo et al. (2019) show that while model configurations with resolutions ranging from 1° to 1/10° degree, can all qualitatively represent broad-scale global patterns of MHWs, modeled MHWs tend to be weaker, longer-lasting, and less frequent than in observations, especially for models with lower resolution. High-resolution, eddy-permitting models perform generally better in representing MHW characteristics, particularly in dynamic regions like western boundary currents, but still exhibit biases (Pilo et al., 2019).

Discrepancies between models and observations are due in part to smoother SST time series and longer autocorrelation times (Cooper et al., 2017) in models (compared to observations), which can reduce short-lived variability and emphasize events of longer duration.

Capotondi, Antonietta, et al. "A global overview of marine heatwaves in a changing climate." Communications Earth & Environment 5.1 (2024): 701.

Cooper, Fenwick C. "Optimisation of an idealised primitive equation ocean model using stochastic parameterization." Ocean Modelling 113 (2017): 187-200.

Forget, G., J.-M. Campin, P. Heimbach, C. N. Hill, R. M. Ponte, and C. Wunsch, 2015: ECCO version 4: an integrated framework for non-linear inverse modeling and global ocean state estimation. Geoscientific Model Development, 8, 3071-3104, http://dx.doi.org/10.5194/gmd-8-3071-2015, http://www.geosci-model-dev.net/8/3071/2015/

Guo, X., Gao, Y., Zhang, S., Cai, W., Chen, D., Leung, L. R., ... & Wu, L. (2024). Intensification of future subsurface marine heatwaves in an eddy-resolving model. Nature Communications, 15(1), 10777.

Pilo, Gabriela S., et al. "Sensitivity of marine heatwave metrics to ocean model resolution." Geophysical Research Letters 46.24 (2019): 14604-14612.

2. *Secondly, in the section discussing the mechanism of global MHWs, the authors conducted a heat budget analysis, but did not present the residual term. Does this mean that ECCO can perform a close heat budget analysis?*

We appreciate the reviewer's comment and the opportunity to clarify this important point.

Although not visible (as orders of magnitude smaller), the residual term is displayed in Figures 11-13. Figure R2 below also shows the residual term corresponding to Figure 7 in the manuscript (which is orders of magnitude smaller than budget terms).

Finally, we included this sentence in Section 3.2 (ECCO ocean heat budget):

"We note that ECCO ocean heat budget closes practically exactly, as residuals are orders of magnitude smaller than the other budget terms."

[Figure]

Fig. R2. Maps representing the heat budget residual for the events average (left), the onset phase (middle) and the decline phase (right), corresponding to Figure 7 in the manuscript.

> ### 3. *Finally, I agree with RC1 that the identification of leading terms is based on predefined thresholds (30%, 16%) that are not explained. The authors should demonstrate this more clearly.*

We appreciate the reviewer's comment regarding the use of predefined thresholds. In the revised manuscript, we implemented a simplified criterion for the identification of leading terms (described later in this response): while the selected threshold is somewhat arbitrary, it helps to summarize our results. Also, we clarify that our results are robust to ∓5% changes in the selected threshold (see figures at the end of this response).

While Figures 9, 11-13 (shown at the end of this response) were updated to reflect the new (simplified) method, results are consistent with the previous version of the plots. We also now show a range for the values in Table 1, to indicate how the percentages change with a threshold value of 25% and 35% instead of 30% (see table at the end of this message). As discussed for the overall classification of leading terms above, the purpose of the table is to summarize which processes are most often leading terms, and the focus is not the exact percentage value reported.

In the revised manuscript we included this text (in the methods section) to introduce the simplified criteria for the identification of leading terms:

"**Towards summarizing our global and regional findings for the leading dynamical processes driving MHWs, we introduce the terms "the" leading term and "a" leading term, defined as follows. For each phase, we identify the budget terms that contribute to it (among forcing, advective convergence, diffusive convergence), then we sort them by the magnitude of the contribution. A term provides "the" leading contribution if it exceeds the next largest term by at least 30%. A term also provides the leading contribution if it is the only process contributing to the phase of interest. If the largest two contributors are both greater than the third by at least 30%, but neither is larger than the other by 30%, then each of the two terms provides "a" leading contribution. The same happens if these two terms are the only contributors and neither is larger than the other by 30%. If none of the terms provides a contribution that is 30% larger than other contributions, all three terms contribute comparably. We note that while the 30% threshold is arbitrary, it serves the purpose of summarizing our results, and visually identifying which processes are most often**

**leading terms. Our findings are robust to ∓5% change in the (30%) threshold percentage used (not shown)."**

Updated Table 1:

| | Terms | NEP (20 events) | | SWP (38 events) | | TASMAN (37 events) | |
|---|---|---|---|---|---|---|---|
| | | **Onset** | **Decline** | **Onset** | **Decline** | **Onset** | **Decline** |
| **"the" leading term** | F | 40-45% | 25-35% | 71% | 29-32% | 38-49% | 38-41% |
| | A | 15-25% | 10-15% | 13% | 18% | 22% | 5% |
| | D | 10% | 20% | 5% | 3% | 0-3% | 11-16% |
| **"a" leading term** | F | 10-25% | 20-30% | 11% | 37-39% | 24-32% | 32-35% |
| | A | 15-25% | 15% | 0% | 13% | 14-19% | 5-8% |
| | D | 5-10% | 25-35% | 11% | 29-32% | 11-19% | 32-35% |

How often each budget term is "the" leading term vs "a" leading term (excluding overlap with "the") across NEP, SWP, and TASMAN regions. A range of percentage values is shown for each case, indicating how percentages change with a threshold value of 25% and 35% instead of 30%, in the definition of "the" vs "a" leading term.

Updated Figure 9:

[Figure]

Figure 9. Same as Figure 8, now for the percentage of times each term is a leading contributor to the MHW (a, c, e) onset and (b, d, f) decline phase. As an example, we include in the count for panel (a) both a case where atmospheric forcing is the only contributor to the onset phase and a case where atmospheric forcing is a leading contributor together with advective and/or diffusive convergence of heat.

Versions of Figure 9 using 25% and 35% (instead of 30%) in the criteria to define "the" versus "a" leading term:

25%

[Figure]

35%

[Figure]

Figure 11 (NEP)

[Figure]

The figure above (and similarly the ones that follow, for other regions) shows the ratio of the total contribution for each budget term during each phase (onset/decline), as well as the tendency (scaled too by the total contribution). In each figure, the top panel is for the onset phase, the bottom one for the decline phase. Each stacked bar represents the relative contribution of each term during that phase, with the total contribution (i.e., the sum of the terms that contribute to that phase) normalized to 1. The black outline over each bar indicates the total temperature tendency during that phase (scaled by the total contribution to that phase), showing agreement with the sum of the individual terms and confirming budget closure. The x-axis labels denote the start date of each MHW event, followed by values in parentheses indicating the event intensity (as average temperature (degC) in the layer used for the OHC estimate), onset/decline duration (in days), and total duration. The letter codes above each bar indicate which term(s) dominated the total temperature tendency during that phase (A = advection, D = diffusion, F = forcing), e.g., "F" is used for cases when the forcing provides the leading contribution, "FA" is used when both forcing and advective convergence provide a leading contribution, "AFD" characterizes cases where advective convergence is larger than forcing and diffusive convergence and forcing is larger than diffusive convergence, yet the difference does not meet the 30% criteria as in the previous two cases. Finally, "~" corresponds to cases where all terms contribute comparably.

**Figure 12 (SWP)**

a)

[Figure]

b)

**Figure 13 (TASMAN)**

a)

[Figure]

b)